# A High Sensitive Flexible Pressure Sensor Designed by Silver Nanowires Embedded in Polyimide (AgNW-PI)

**DOI:** 10.3390/mi10030206

**Published:** 2019-03-24

**Authors:** Hongfang Li, Guifu Ding, Zhuoqing Yang

**Affiliations:** National Key Laboratory of Science and Technology on Micro/Nano Fabrication, School of Electronic Information and Electrical Engineering, Shanghai Jiao Tong University, Shanghai 200240, China; hfli2014@sjtu.edu.cn (H.L.); yzhuoqing@sjtu.edu.cn (Z.Y.)

**Keywords:** flexible pressure sensor, high sensitivity, silver nanowires polyimide (AgNW-PI), selective wet etching

## Abstract

Silver nanowires (AgNW) have excellent electrical conductivity, transparency, and flexing endurance, and are broadly used in flexible electrodes and flexible sensors. This study mixed the silver nanowires and polyimide (PI) polymer using an in situ synthesis method, effectively reducing the problem of silver nanowires falling off the substrate. The selective wet etching method was firstly used to process the surface of AgNW-PI films, greatly enhancing the surface conductivity of AgNW-PI films. A flexible pressure sensor with high sensitivity was designed with two face-to-face AgNW-PI ultrathin layers. The experimental results show that our sensor presented a high sensitivity of about 1.3294 kPa^−1^ under a pressure of about 600 Pa, and when pressure continued to increase, the sensitivity decreased rapidly and reached saturation. Our flexible pressure sensor has the properties of low cost, high sensitivity, excellent repeatability, durability, and can detect various types of mechanical forces which could be utilized for flexible electronics.

## 1. Introduction

Flexible pressure sensors are widely used in intelligent clothing, intelligent movement, robot “skin”, and more. In order to meet the requirements of flexible pressure sensing, thin, transparent, flexible, and good electrical performance become the key indicators of the sensor’s quality.

Traditional pressure sensors, for example, silicon pressure sensors [1,2], have a good piezoresistive effect and good linear properties. Nevertheless, the silicon is hard and brittle, which is difficult for flexible devices. Silicone rubber [3,4], polydimethylsiloxane [5,6,7], and polyimide [8] have been widely used in the manufacture of flexible devices. Silicone rubber and polydimethylsiloxane are not compatible with micromachining technologies such as lithography. Polyimide (PI), with its excellent heat-resistance, fine strength, rigidity, and compatibility with microfabrication processes, is one of the best organic polymer materials available with high comprehensive performance [9]. Up to now, PI has widely served as a structural and functional material in microelectronic devices [10].

Most scholars have attempted to synthesize functional materials in order to simultaneously attain sensor flexibility and conductivity. The active materials with excellent conductivity, like metal particles [11], metal nanowires [12], carbon black [4,13,14], graphene [15], and carbon nanotubes [16,17,18,19,20], are usually combined with a flexible substrate material, creating functional materials with excellent conductivity and flexibility. 

Silver nanowires (AgNWs), as one-dimensional metal nanowires, have been widely used as flexible sensors due to their excellent conductivity and flexing endurance. How to uniformly disperse the Ag nanowires, and improve the adhesion strength of Ag nanowires to a polymer substrate, are the main pressing questions yet to be answered. Amjadi et al. [21] reported a highly stretchable and sensitive strain sensor based on sandwich-structured polydimethylsiloxane (PDMS)/AgNW/PDMS nanocomposites, with the Ag nanowire solution spread onto the patterned glass slide with polyimide tape. The spreading process was easy to perform, but the Ag nanowires were not easy to disperse uniformly, and it was possible for them to aggregate before drying. Lagrange. et al. [22] demonstrated that Ag nanowires could be spin-coated at low rpm. Using this method, ultrathin Ag nanowire films with uniform dispersion can be obtained. However, a major drawback of this method is the poor adhesion strength between the Ag nanowire film and the substrate, which limits their fine-pitch pattern ability and holds back their practical usage. Nam et al. [23] adopted curable polymer of Norland Optical Adhesive NOA 63 liquid photopolymer photocured by ultraviolet exposure to transfer AgNWs from the mother substrate. The merit of this transfer method is that the adhesion of silver nanowires to polymers is greatly improved, and the drawback is that it is incompatible with photolithography.

This study presents a flexible pressure sensor fabricated by two pieces of face-to-face AgNW-PI layers with the AgNW-PI layers used as pressure-sensing elements. AgNW-PI composite was formed by mixing Ag nanowires with PI polymer, which avoided the problem of poor adhesion. This pressure sensor has the advantages of high sensitivity, flexibility, and low cost. The remainder of this study is organized as follows: Section 2 is the design and operating principles; Section 3 presents the materials and methods; Section 4 is the results and discussion; last, the conclusion is summarized.

## 2. Design and Operational Principles

The proposed flexible sensor is shown in Figure 1a. The sensor includes two face-to-face AgNW-PI layers which functioned as pressure-sensing layers, Ni electrodes fabricated by electroplating, PI layers used as substrate, and polyvinylchloride (PVC) film used as the substrate and encapsulation layer. Figure 1b demonstrates the real fabricated sensor with an area of 3 × 3 cm^2^. The sensor bends easily by hand, which demonstrates that the sensor has good flexibility. The enlarged view of the interface of face-to-face AgNW-PI layers is presented in Figure 1c. On the surface of the film, there are many silver nanowires with one end embedded in the polymer and the other end exposed on the surface.

As shown in Figure 2, the sensing mechanism of this flexible pressure sensor with two face-to-face AgNW-PI films with surface wet etching is that when pressure is not exerted on the device, the Ag nanowires on the top layer are partly and randomly in contact with those on the bottom layer. However, when we touch the sensor or apply pressure onto it, more Ag nanowires on the two layers come into contact with each other, resulting in an increase in the number of possible conduction pathways, leading to a sharp drop in resistance. Correspondingly, the sensitivity of the sensor will be high. As the pressure increases to a certain extent, the conduction pathways reach saturation. As the pressure continues to increase, the increase of Ag nanowire networks in the membrane prompts a slow change of resistance.

## 3. Materials and Methods

### 3.1. Materials and Fabrication Process

The high purity Ag nanowires (AgNWs) (average length of 20 μm and average diameter of 120 nm) suspended in alcohol at a concentration of 20 mg/mL were purchased from the XFNANO company (XFNANO Materials Tech Co., Ltd., Nanjing, China). Polyimide (PI) with an absolute viscosity of 1100–1200 mPa·s was purchased from POME Sci-tech Co., Ltd. (Beijing, China).

In this study, the Ag nanowires were mixed with PI to form a composite material with 2% mass concentration. Next, the composite was degassed in the vacuum oven for 30 min. Then, the prepared composites were removed from the vacuum oven and set aside. By spin-coating, selective wet etching, sputtering, photolithography, electroplating technology, PI worked as a sacrificial layer, and AgNW-PI films were transferred onto the flexible substrate. The specific preparation process is presented as follows (Figure 3):(a)The glass substrate was cleaned with alkaline solution, acid solution, and deionized water. The PI film was pasted and a Cr/Cu seed layer was applied by magnetron sputtering for electroplating.(b)The substrate was spin-coated with positive photoresist (AZ 4620) with 10 µm thickness. The photoresist was patterned by lithography and developed. A thin layer of Ni metal (the area was about 2 × 2 cm^2^) was electroplated to 3–5 µm thickness. This layer of Ni metal was used as the electrode.(c)The excess photoresist and Cr/Cu seed layer were removed by alkaline solution. The surface was etched with O_2_ plasma for 2 min to remove organic residue.(d)Through several experiments, the spin-coating machine was set to a speed of 1500 r/min. Before the official spin-coating of AgNW-PI composite, a few drops of AgNW-PI composite were placed on the center of the substrate, and the rotation was gradually accelerated to ensure uniform coating of AgNW-PI composite onto the substrate, which enhances the adhesion of the AgNW-PI composite to the substrate. The AgNW-PI composite was then spin-coated for 25 s. The method of step imidization was used, baking at 80 °C for 30 min, and then baking at 110 °C for 1 h. The obtained AgNW-PI film had a thickness of about 20 μm and did not wrinkle, and the imidized structure was suitable for further use.(e)A 30 µm-thick positive photoresist (AZ 4903) was spin-coated and was patterned by lithography. The photoresist and AgNW-PI composite were developed.(f)The photoresist was removed using acetone to prevent removal of the remaining AgNW-PI composite.(g)The AgNW-PI film was wet etched by NaHCO_3_ solution with 1% mass concentration for 5 min in order to expose the Ag nanowires.(h)The PI film was peeled off from the glass substrate and pasted onto the thin flexible PVC substrate. Then, the conductive copper wires were welded on the Ni electrodes by a point welding machine and acrylic modified epoxy adhesive was utilized to reinforce the welding.(i)Two pieces of the abovementioned AgNW-PI films on the PVC flexible film were stuck face-to-face using double-sided adhesive PI tape and packaged using flexible adhesive tape to obtain a flexible pressure sensor.

### 3.2. Experimental Setup

As shown in Figure 4, a self-assembly measurement system was designed to measure different characteristics of the flexible pressure sensor, such as sensibility, time-resolved response, and mechanical response. The measuring system included a pressure device (bonding tester), a data acquisition device (digital multimeter), and a data recording and analysis device (computer controller). A bonding tester (PTR-1101, RHESCA, Japan) consists of a precision pressure sensor, a sample holder, and a pressure head. This equipment was used to provide external normal force of up to 200 N (the measurement accuracy was 0.1 N) and the sensor was fixed on the sample holder below the pressure head. The resistance was tested and recorded by a digital multimeter (Agilment 34465A,6_1__/2_, Keysight Technology, Santa Rosa, CA, USA). The computer controller performed the mechanical and electrical signals of reception, processing, displaying, and recording.

## 4. Results and Discussion

### 4.1. Surface Morphology

The scanning electron micrographs (SEM) of the surface of prepared AgNW-PI films without wet etching and with wet etching are demonstrated in Figure 5. For the surface without wet etching (Figure 5a), on the surface of the film, nearly all the Ag nanowires were wrapped in the PI polymer, which resulted in the high surface resistance of AgNW-PI composite film. After testing, the surface resistance was infinite with the digital multimeter. This was mainly due to the surface effect of nanostructured materials, Ag nanowires have high surface-free energy that is larger than the surface tension of PI polymer, which resulted in the Ag nanowires being entirely wetted by PI polymer. In order to improve this situation, a wet etching method was utilized to etch the surface of AgNW-PI film. In this method, the etching solution was NaHCO_3_ solution with 1% mass concentration. The AgNW-PI film was etched in 5 min and resulted in the SEM surface morphology demonstrated in Figure 5b. For most of the Ag nanowires, some parts was exposed on the surface of the AgNW-PI film, while other parts remained embedded in the PI matrix. The embedded parts of Ag nanowires enhanced the adhesion strength of AgNWs and PI polymer, which effectively reduced the possibility of AgNWs falling off the surface compared with directly spin-coated Ag nanowires [22]. Also, we can see that Ag nanowires are uniformly dispersed in the PI polymer and the uncovered Ag nanowires are jointed and cross with each other, giving rise to networks. When we apply pressure on the face-to-face AgNW-PI sensor, the exposed Ag nanowires on the top and bottom layer connect with each other and numerous conductive pathways are created, which will greatly decrease the resistance of the device.

### 4.2. Sensitivity of the Flexible Pressure Sensor

Sensitivity is one of the most important indexes for measuring a sensor’s quality. Here, we defined the sensitivity of our sensor as
ΔR = R − RoS = (ΔR/Ro)/ΔP,where Ro is the original resistance under no pressure; R is the measured resistance when applied pressure; ΔR is the relative resistance change obtained by R-Ro; ΔP is the change in applied pressure.

As shown in Figure 6, the sensor presents nearly linear negative piezoresistive effect. When the applied pressure was smaller than 600 Pa, the red fitted line gave a high sensitivity of about 1.3294 kPa^−1^, also demonstrated by the inset graph. The sensitivity of this sensor was higher than the graphite nanoplatelet carbon nanotube hybrid/PDMS composite pressure sensor (~0.6 kPa^−1^) [24], the horizontally oriented carbon nanotube network pressure sensor (1.68%/kPa) [25], among others. With the increased application of pressure, the sensitivity decreased to 0.0235% kPa^−1^. This may mainly be due to the fact that under no pressure, the two face-to-face AgNW-PI layers are not, or nearly not, in contact, resulting in very large resistance. When a small amount of pressure was applied to the sensor, the two layers of AgNW-PI touch each other and most Ag nanowires on the surface came into contact, forming conductive pathways which led to the drastic reduction in resistance. In addition, the sensitivity was high. When the applied pressure reaches saturation, the change in resistance slows. When the pressure increased further, the resistance changed mainly through the Ag nanowire networks in the AgNW-PI film, which resulted in a small change in resistivity. Hence, this sensor could be used to detect small amounts of pressure.

### 4.3. Resistance Response under Constant Pressure

From Figure 7, we can see that under different constant pressures of 3.97, 7.94, and 11.91 kPa, the resistance change rate increased with the increase of the constant pressure. Also, the resistance change rates were almost equal under the same constant pressure at different cycles, and once the pressure was released, the resistance quickly returned to the original value. These observations demonstrate the good stability of the sensor.

### 4.4. Resistance Response under Different Types of Mechanical Forces

The sensor was used to detect the different types of mechanical forces of touching, bending, and torsional forces in Figure 8. When we used a finger to touch the sensor, the resistance change rate was about 0.5, and when we moved the finger, the resistance quickly responded to the original value (Figure 8a). Similarly, when bending load and torsion load was applied to the sensor, the resistance change rate respectively reached to about 0.8 and 1.0 (Figure 8b,c). However, when the pressure was released, the deformations of the top and bottom face-to-face AgNW-PI layers were different, which resulted in the two layers separating, simultaneously, and the resistance of the sensor became larger than the original resistance. To address this problem, we could change the packaging method.

### 4.5. Dynamic Test

The experimental setup used for characterization of dynamic performance is shown in Figure 9. The flexible pressure sensor was mounted onto a shaker to observe the output resistance response to input vibrations. The fixed mount was pressed onto the sensor. The shaker is driven by the data acquisition (DAQ) card of NI USB-6356 through a power amplifier. The resistance was measured by the digital multimeter (Keysight 34465A, Keysight Technology), and was recorded by the BenchVue software. In the experiment, the sinusoidal excitation frequency was 1 Hz at 1 g excitation level.

To detect the dynamic synchrony of the sensor between the cyclic pressure and corresponding resistive change, the multicycle piezoresistive tests under sinusoidal pressure were done to the sensor in Figure 10. Excellent repeatability and long-term piezoresistive behavior of the sensor was achieved in the following 500 cycles. However, the stability of the sensor remains to be improved.

## 5. Conclusions

Using in situ synthesis and selective wet etching method, we overcame the problems of Ag nanowire dispersion and adhesion strength with polymers, and successfully obtained AgNW-PI film. The flexible pressure sensor designed with two face-to-face AgNW-PI films demonstrated that it had the properties of high sensitivity of about 1.3294 kPa^−1^ under 600 Pa, easy fabrication, low cost, compatibility with MEMS technology, excellent repeatability, and durability. The findings of this paper provide foundations for portable wearable devices, robots, and applications in the biomedicine industry. Moreover, we will reduce the size of the device and design the sensor array to achieve a higher sensitivity and cover large areas.

## Figures and Tables

**Figure 1 micromachines-10-00206-f001:**
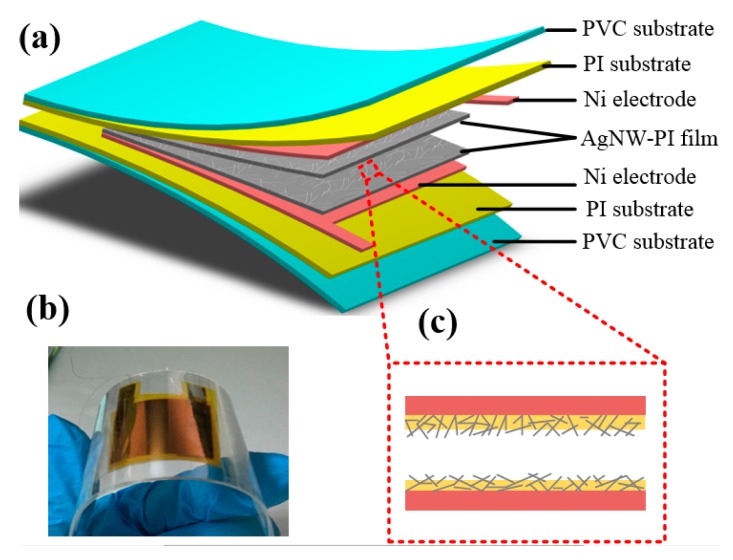
(**a**) The proposed flexible sensor with two face-to-face silver nanowires polyimide (AgNW-PI) layers, Ni electrodes, PI flexible substrates and polyvinylchloride (PVC) flexible encapsulation layer; (**b**) the real physical photo of the sensor; (**c**) the schematic interface of the two face-to-face AgNW-PI layers.

**Figure 2 micromachines-10-00206-f002:**
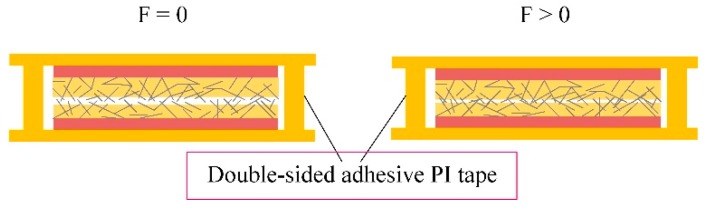
The schematic diagram of the sensing mechanism.

**Figure 3 micromachines-10-00206-f003:**
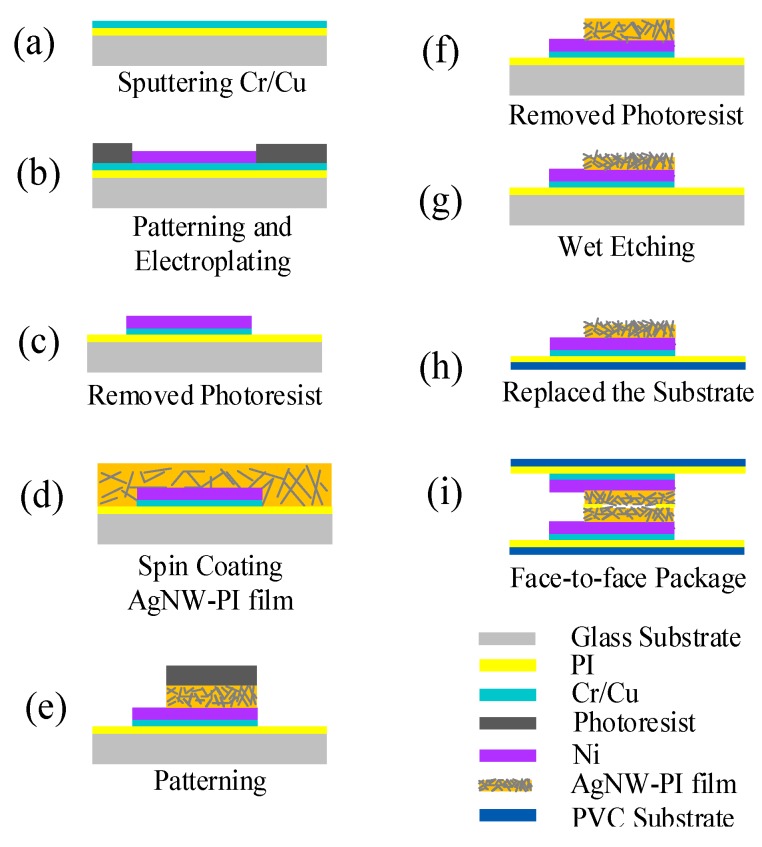
The schematic fabrication process of the flexible pressure sensor. (**a**) Sputtering Cr/Cu seed layer; (**b**) patterning and electroplating; (**c**) removed photoresist; (**d**) spin coating AgNW-PI film; (**e**) pattering; (**f**) removed photoresist; (**g**) wet etching; (**h**) replaced the substrate and (**i**) face-to face package.

**Figure 4 micromachines-10-00206-f004:**
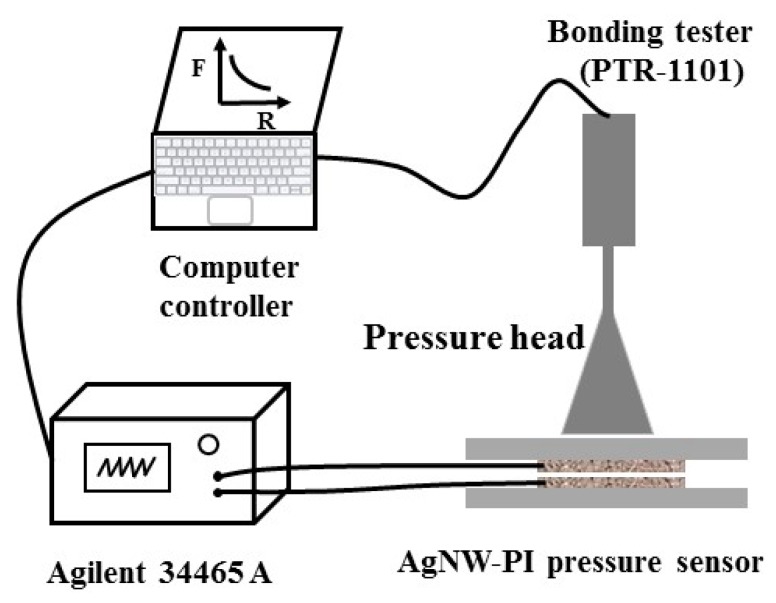
The schematic illustration of the testing setup.

**Figure 5 micromachines-10-00206-f005:**
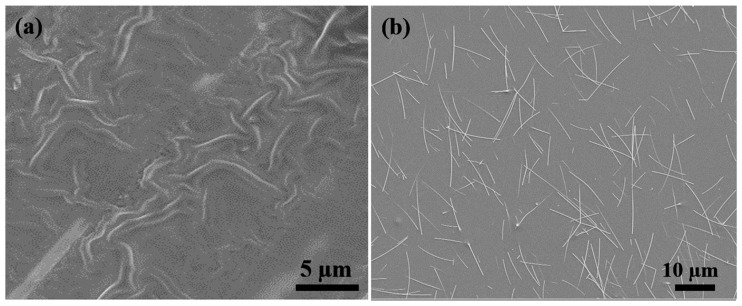
The surface morphology of AgNW-PI film by SEM: (**a**) the original surface without wet etching; (**b**) the surface through wet etching.

**Figure 6 micromachines-10-00206-f006:**
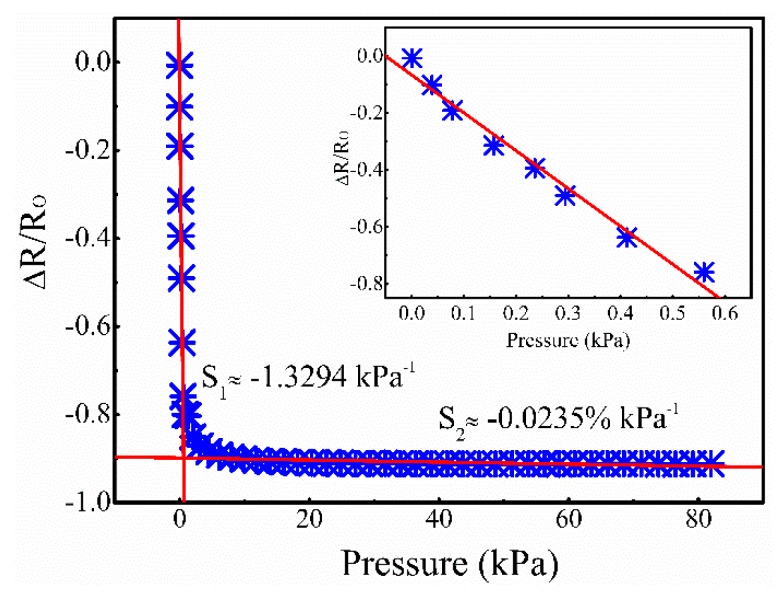
The resistance response to various applied pressures. The red line is the fitted lines and the inset graph is the part enlargement (*p* < 600 Pa).

**Figure 7 micromachines-10-00206-f007:**
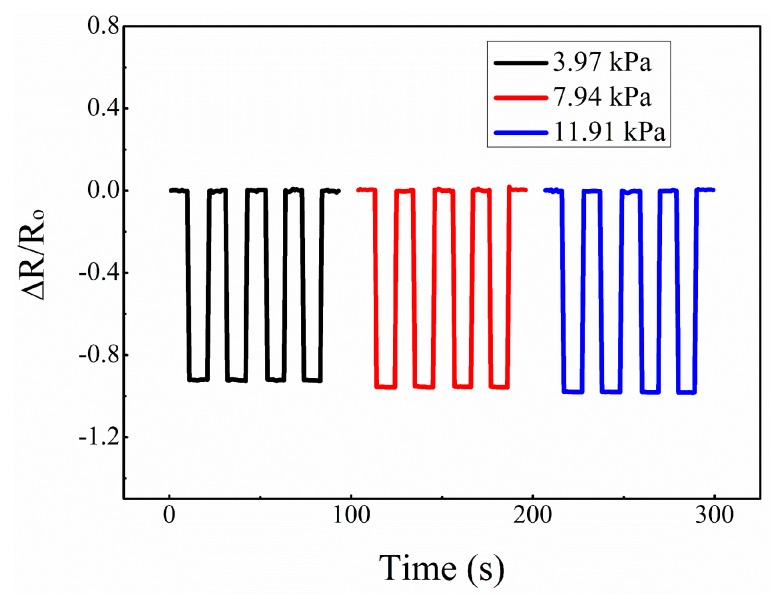
Resistance response under constant pressure.

**Figure 8 micromachines-10-00206-f008:**
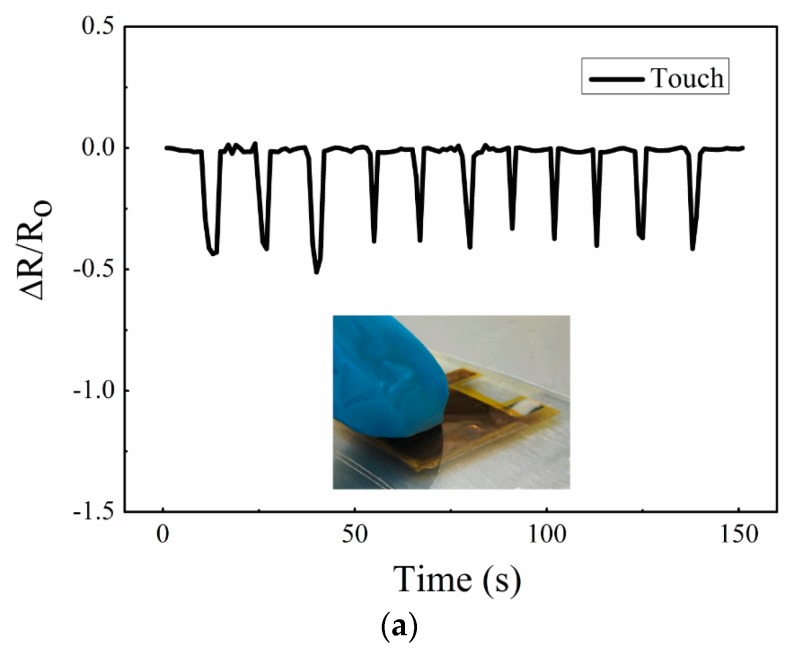
Resistance response under different types of mechanical forces: (**a**) under finger touch; (**b**) under bending force; (**c**) under torsion pressure.

**Figure 9 micromachines-10-00206-f009:**
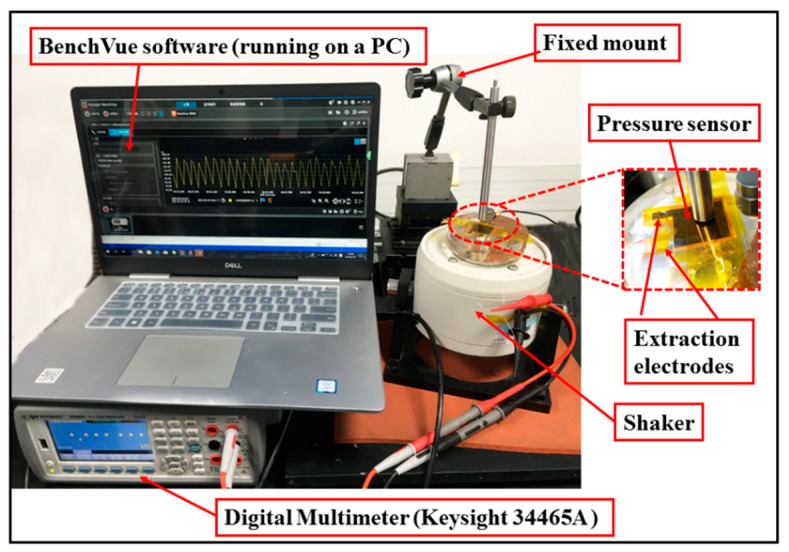
The experimental setup system for dynamic test.

**Figure 10 micromachines-10-00206-f010:**
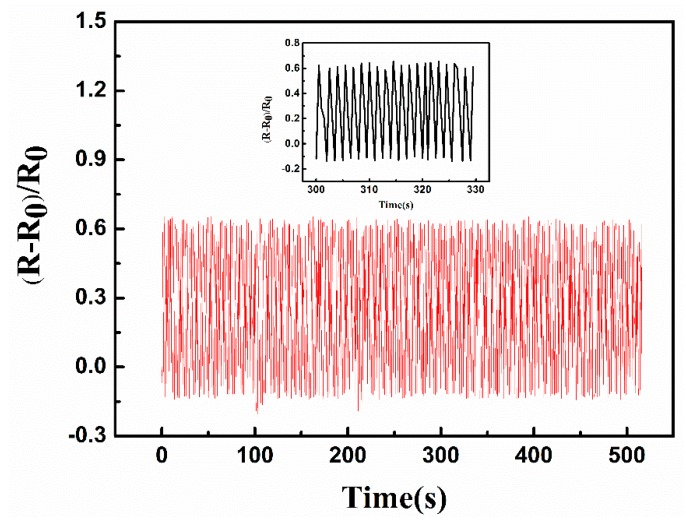
Repeatability testing for about 500 cycles of the sensor.

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
