# Peer review of "A High Sensitive Flexible Pressure Sensor Designed by Silver Nanowires Embedded in Polyimide (AgNW-PI)"

_micromachines, 2019, doi:10.3390/mi10030206_

Round 1

Reviewer 1 Report

In the introduction, there is often a mixed discussion about strain sensors and presser sensors which are not the same. This makes it hard for the reader to follow the significance of the presented work compared to other relevant studies.

Moreover, most of the applications presented in the article are related to health monitoring (such as heart beat measurement, etc). However, it is not clear how the presented device can be implemented for such applications. The instrumentation presented in one of the figures, shows that the system might be complicated for the proposed applications. How can it be compared with the current sensitive systems available for such proposes?

In addition to electrical measurement, optical measurement is another method for force sensing and it should be mentioned in the introduction. See for example:

DOI: 10.1002/smll.201801187
DOI: 10.1021/acsnano.7b08290

Have the authors assessed the effect of thicknesses (multiple layers) on the performance of the sensors?

Does this device have a break down point?

It is important to assess the sensor over multiple cycles until decay. How many cycles the sensor can stand?

What are the optimal voltage and current for the electrical readout?

Author Response

Dear Micromachines Referee,

Re: "A High Sensitive Flexible Pressure Sensor Designed by Silver Nanowires Embedded in Polyimide (AgNW-PI)" by Hongfang Li; Guifu Ding, Zhuoqing Yang

Article reference: Micromachines-458934

To begin with, we would like to express our thanks to the two referees for their careful review and constructive suggestions with regard to our manuscript. After getting the referees’ comments, we studied it carefully and tried our best to revise and improve the manuscript according to the comments. We gave point-by-point responses to the referees’ comments, as detailed below. In the revised manuscript, all textual revisions are marked in red. A clean final version of our revised manuscript is uploaded as well.

Thanks for all the help!

Yours sincerely,

Hongfang Li

Referee: 1

COMMENTS TO THE AUTHOR(S)

1.        In the introduction, there is often a mixed discussion about strain sensors and presser sensors which are not the same. This makes it hard for the reader to follow the significance of the presented work compared to other relevant studies.

Response:

We greatly appreciate the reviewer’s valuable comments and suggestions. Thanks for this good suggestion of the reviewer.

We have revised the Introduction as detailed below:

Flexible pressure sensors are widely used in intelligent clothing, intelligent movement, robot “skin” and other aspects. In order to meet the requirements of flexible pressure sensing, thin, transparent, flexible and good electrical performance become the key indicators.

Traditional pressure sensors, for example silicon pressure sensors [1, 2] have good piezoresistive effect and good linear property. Nevertheless, the silicon is hard and brittle, which is difficult for flexible devices. Silicone rubber[3, 4][Wang, 2009 #77], polydimethylsiloxane[5-7] and polyimide[8] have been widely used in the manufacture of flexible devices. Silicone rubber and polydimethylsiloxane are not compatible with micro-machining technologies such as lithography. Polyimide (PI) with excellent heat-resistance, fine strength and rigidity and compatible with microfabrication process, which is one of the best organic polymer materials with high comprehensive performance [9]. Now, PI has been widely served as a structural and functional material in microelectronic devices [10].

Most scholars attempt to synthesis some functional materials in order to attain the sensor’s flexibility and conductivity simultaneously. The active materials with excellent conductivity like metal particles [11], metal nanowires [12], carbon black [13-15], graphene [16], and carbon nanotubes [17-21] are usually combined with flexible substrate material, obtaining the functional materials with excellent electricity and flexibility.

Silver nanowires (AgNWs), as one-dimensional metal nanowires, with excellent conductivity and flexing endurance, have been widely used for flexible sensors. Uniform dispersion of Ag nanowires and the adhesion strength of Ag nanowires to polymer substrate are the main urgent questions to solve. Morteza Amjadi et al [22] reported a highly stretchable and sensitive strain sensor based on the sandwich-structured PDMS/AgNW/PDMS nanocomposites and the Ag nanowires solution was spreaded on the patterned glass slide with Polyimide tape. The process of spreading method is easy to operate, but the Ag nanowires were not easy to disperse uniformly and possible to reunite before drying. M. Lagrange. et al [23] presented that Ag nanowires could be spin-coated with low RPM. By this method, Ag nanowire films with uniform dispersion and ultrathin thickness can be obtained. However, a major drawback of this method is poor adhesion strength between Ag nanowires film and the substrate, which limits their fine-pitch pattern ability and holds back their practical usage. Sanggil Nam et al [24] adopted NOA 63 liquid photopolymer photo-cured by UV exposure to transfer AgNWs from the mother substrate. The merit of transfer method is that the adhesion of silver nanowires to polymers is greatly improved, and the defect is that it is incompatible with photolithography.

This study presents a flexible pressure sensor fabricated by two pieces of face-to-face AgNW-PI layers and the AgNW-PI layers used as pressure sensing element. AgNW-PI composite is formed by mixing Ag nanowires with PI polymer, which avoided the problem of poor adhesion. This pressure sensor features advantages of high sensitivity, flexibility and low cost. The remainder organizations of this study are as follows: section 2 is the design and operating principles; section 3 presents the materials and methods; section 4 is the results and discussion; last, the conclusion is summarized.

2.        Moreover, most of the applications presented in the article are related to health monitoring (such as heart beat measurement, etc). However, it is not clear how the presented device can be implemented for such applications. The instrumentation presented in one of the figures, shows that the system might be complicated for the proposed applications. How can it be compared with the current sensitive systems available for such proposes?

Response:

Now the article is only a basic research, which has not reached the requirements of wearable devices. The previous description is not accurate, so the introduction part has been modified as the answer to question 1.

3.        In addition to electrical measurement, optical measurement is another method for force sensing and it should be mentioned in the introduction. See for example:

DOI: 10.1002/smll.201801187
DOI: 10.1021/acsnano.7b08290

Response:

Optical measurement is indeed a good method of force sensing, but due to the limited conditions in our laboratory, this paper adopts the method of resistance measurement.

Your papers have prepared flexible sensors with PDMS, which is of great significance to our paper.

Silicone rubber[3, 4][Wang, 2009 #77], polydimethylsiloxane[5-7] and polyimide[8] have been widely used in the manufacture of flexible devices. Silicone rubber and polydimethylsiloxane are not compatible with micro-machining technologies such as lithography.

References

6.    Tiefenauer R. F.; Dalgaty T.; Keplinger T.; Tian T.; Shih C. J.; Vörös J. and Aramesh M. Monolayer Graphene Coupled to a Flexible Plasmonic Nanograting for Ultrasensitive Strain Monitoring. Small. 2018, 14, pp. 1-7.

7.    Tiefenauer R. F.; Tybrandt K.; Aramesh M. and Vörös J. Fast and Versatile Multiscale Patterning by Combining Template-Stripping with Nanotransfer Printing. Acs Nano. 2018, 12, pp. 2514-2520.

4.        Have the authors assessed the effect of thicknesses (multiple layers) on the performance of the sensors?

Response:

Thank you for your constructive suggestion! The paper focuses on the preparation and test of the structure. Thickness and the concentration of silver nanowires doped in polyimide may affect the performance of the sensor, which will be studied in the future research.

5.        Does this device have a break down point?

Response:

As the AgNW-PI film surface was etched by wet etching, large number of silver nanowires were exposed on the surface, resulting in low resistance. So, the resistance measured with this device is continuous, within the range of the device, and there is no break point.

6.        It is important to assess the sensor over multiple cycles until decay. How many cycles the sensor can stand?

Response:

We take your advice and test on the device for about 500 cycles as follows:

4.5 Repeatability test

Figure 8. Device diagram for repeatability test.

Figure 9. Repeatability testing for about 500 cycles of the sensor.

7.        What are the optimal voltage and current for the electrical readout?

Response:

The device used to test the resistance was KEYSIGHT 34465A Digital Multimeter, which had a resolution of 61/2. BenchVue Sofware (running on a PC) is a device-supplied software that allows users to easily connect instruments, record data, and obtain measurement results without programming. Quickly record and export data to common tools such as Microsoft Excel, Microsoft Word, and MATLAB for archiving or further analysis. The resistance value measured by two-wire resistance in the experiment.

Reviewer 2 Report

The authors presented a high sensitive flexible pressure sensor by silver nanowires embedded in Polyimide. The presented results are valuable. However, following comments are suggested

1. In the report, the authors use photolithography to define electrodes and electroplating to deposit Ni films. And it is claimed that the Ag nanowires on the top layer are not or nearly not contact with those on the down layer.  But the authors did not explain what is the electrode size and how to align these two layers face-to-face for preventing from direct contact. It seems a tricky process and will influence the sensitivity.

2. The sentence of LINE 106 to 107 should be revised.

3. The sentence of LINE 137 to 139 should be revised.

4. In Fig.5, the authors shows the resistance response to various applied pressures. But on LINE 78~79, the authors claimed the initial resistance is infinite. Does it means R0 is infinite? If R0 is infinite, it is difficult to understand how to calculate deltaR. The authors should explain more clearly about R0. Otherwise, the presented results seems make no sense.

Author Response

Dear Micromachines Referee,

Re: "A High Sensitive Flexible Pressure Sensor Designed by Silver Nanowires Embedded in Polyimide (AgNW-PI)" by Hongfang Li; Guifu Ding, Zhuoqing Yang

Article reference: Micromachines-458934

To begin with, we would like to express our thanks to the two referees for their careful review and constructive suggestions with regard to our manuscript. After getting the referees’ comments, we studied it carefully and tried our best to revise and improve the manuscript according to the comments. We gave point-by-point responses to the referees’ comments, as detailed below. In the revised manuscript, all textual revisions are marked in red. A clean final version of our revised manuscript is uploaded as well.

Thanks for all the help!

Yours sincerely,

Hongfang Li

Referee: 2

COMMENTS TO THE AUTHOR(S)

The authors presented a high sensitive flexible pressure sensor by silver nanowires embedded in Polyimide. The presented results are valuable. However, following comments are suggested:

1.        In the report, the authors use photolithography to define electrodes and electroplating to deposit Ni films. And it is claimed that the Ag nanowires on the top layer are not or nearly not contact with those on the down layer. But the authors did not explain what is the electrode size and how to align these two layers face-to-face for preventing from direct contact. It seems a tricky process and will influence the sensitivity.

Response:

We greatly appreciate the reviewer’s valuable comments and suggestions.

The sentences of LINE 76 to 80 (The possible sensing mechanism) are really error descriptions of my negligence. The original sentences have been changed as follows:

The possible sensing mechanism of this flexible pressure sensor with two face-to-face AgNW-PI films with surface wet etching is that when the device is not exerted pressure or not bend, the Ag nanowires on the top layer are partly and randomly contact with those on the down layer. However, when we touch the sensor or apply a pressure on it, the more Ag nanowires on the two layers contact with each other, resulting in considerable conduct paths, leading to a sharp drop in resistance.

2.      The sentence of LINE 106 to 107 should be revised.

Response:

Thank you very much for your rigorous academic attitude. The sentences of LINE 106 to 107 have been revised as follows:

The scanning electron micrographs (SEM) of the surface of prepared AgNW-PI films without wet etching and with wet etching respectively demonstrate in Figure 4. For the surface without wet etching (Figure 4(a)), on the surface of the film, nearly all of the Ag nanowires are wrapped in the PI polymer, resulting in the high surface resistance of AgNW-PI composite film. After testing, the surface resistance is infinite with the digital multimeter.

3.      The sentence of LINE 137 to 139 should be revised.

Response:

The sentences of LINE 137 to 139 have been revised as follows:

Through several experiments, the spin coating machine was set to a speed of 1500 r/min. Before the official spin-coating of AgNW-PI composite, a few drops of AgNW-PI composite were placed on the center of the substrate, and the rotation was gradually accelerated to make AgNW-PI composite uniformly coat on the substrate, which enhances the adhesion of the AgNW-PI composite to the substrate. The AgNW-PI composite was then spin coated for 25 s. The method of step imidization was used, baking at 80 °C for 30 min, and then baking at 110 °C for 1 h. The obtained AgNW-PI film had a thickness of about 20 μm and did not wrinkle, and the imidized structure was very good.

4.        In Fig.5, the authors shows the resistance response to various applied pressures. But on LINE 78~79, the authors claimed the initial resistance is infinite. Does it means R0 is infinite? If R0 is infinite, it is difficult to understand how to calculate deltaR. The authors should explain more clearly about R0. Otherwise, the presented results seems make no sense.

Response:

For the surface of AgNW-PI film without wet etching, the silver nanowires are wrapped in the PI, so, the resistance is infinite. However, for the sensor with two face-to-face AgNW-PI films with wet etching, due to the double-sided PI tapes (about 100um thickness) stick the two face-to-face AgNW-PI films under an optical microscope, the upper and down AgNW-PI layers have a tiny gap. The partial exposed silver nanowires on the upper and down layers randomly contact with each other, resulting a limited and relatively resistance (the actual resistance in the experiment is less than 1000Ωand the resistance varies with the density of silver nanowires in the PI). So, the Fig.5 is correct. The line 78-79 is my unclear description and has devised as follows:

 The possible sensing mechanism of this flexible pressure sensor with two face-to-face AgNW-PI films with surface wet etching is that when the device is not exerted pressure or not bend, the Ag nanowires on the top layer are partly and randomly contact with those on the down layer. However, when we touch the sensor or apply a pressure on it, the more Ag nanowires on the two layers contact with each other, resulting in considerable conduct paths, leading to a sharp drop in resistance.

Round 2

Reviewer 1 Report

The authors have addressed my previous concerns in the previous review, therefore the paper should be published.